# Recent Applications of Flavin-Dependent Monooxygenases in Biosynthesis, Pharmaceutical Development, and Environmental Science

**Yuze Guan and Xi Chen *** 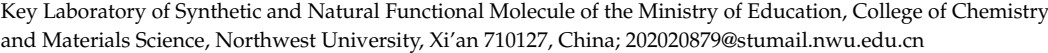

Key Laboratory of Synthetic and Natural Functional Molecule of the Ministry of Education, College of Chemistry and Materials Science, Northwest University, Xi'an 710127, China; 202020879@stumail.nwu.edu.cn
* Correspondence: xchen@nwu.edu.cn

**Abstract:** Flavin-dependent monooxygenases (FMOs) have raised substantial interest as catalysts in monooxygenation reactions, impacting diverse fields such as drug metabolism, environmental studies, and natural product synthesis. Their application in biocatalysis boasts several advantages over conventional chemical catalysis, such as heightened selectivity, safety, sustainability, and eco-friendliness. In the realm of biomedicine, FMOs are pivotal in antibiotic research, significantly influencing the behavior of natural products, antimicrobial agents, and the pathways critical to drug synthesis They are also underscored as potential pharmaceutical targets, pivotal in opposing disease progression and viable for therapeutic intervention. Additionally, FMOs play a substantial role in environmental science, especially in pesticide processing and in preserving plant vitality. Their involvement in the biosynthesis of compounds like polyethers, tropolones, and ω-hydroxy fatty acids, with remarkable regio- and stereoselectivity, renders them indispensable in drug discovery and development. As our comprehension of FMOs' catalytic mechanisms and structures advances, through the use of cutting-edge biotechnologies like computational design and directed evolution, FMOs are poised to occupy an increasingly significant role in both scientific exploration and industrial applications.

**Keywords:** flavin-dependent monooxygenases (FMOs); natural product biosynthesis; biocatalysis; pharmaceutical development; heteroatom hydroxylation; Baeyer–Villiger oxidation

## 1. Introduction

Flavin-dependent enzymes are a class of enzymes that play a crucial role in biochemical reactions, catalyzing a variety of chemical reactions using flavins as cofactors. Flavins, including flavin adenine dinucleotide (FAD) and flavin mononucleotide (FMN), are water-soluble derivatives of vitamin B2, ubiquitously present in organisms. These cofactors act as catalysts in numerous biochemical processes, especially in redox reactions, and are essential for maintaining the normal metabolism in living organisms [1]. Among the diverse families of flavin-dependent enzymes, flavin-dependent monooxygenases (FMOs) are particularly significant. These enzymes catalyze monooxygenation reactions, where a single oxygen atom is inserted into an organic substrate, utilizing the unique redox properties of flavin cofactors. Biologists and chemists have begun to exploit these incredible catalysts in various fields, aiming to provide their potential applications in biocatalysis across diverse industries in the future.

Distinct from traditional chemical catalysts, biocatalysts offer unique advantages in terms of high selectivity, safety, sustainability, and eco-friendliness [2]. For flavin-dependent monooxygenases (FMOs), the mediated redox reactions utilize molecular oxygen as a pure chemical oxidant, with one oxygen atom being incorporated into the substrate, and the other forming water. This process circumvents the use of challenging and hazardous oxidants. As biocatalysts, FMOs can operate in aqueous environments without the need

for extreme temperatures or pressures, simplifying and enhancing the practicality of biocatalytic processes. Despite these advantages, the widespread application of biocatalysis across various fields faces significant challenges, such as the limited number of characterized enzymes, difficulties in enzyme acquisition, and narrow substrate scope. However, with advances in molecular and structural biology, research on FMOs is intensifying, revealing their complex regulatory mechanisms and interactions with biomolecules [3,4]. The ongoing development of biocatalysis has led to an increase in the number of commercial enzymes, and technologies for enzyme modification and evolution have been established to expand substrate scope and enhance enzyme stability [5].

Years of research elucidated the mechanistic framework of flavin-dependent monooxygenases. As illustrated in Figure 1, flavins undergo a series of oxidative epoxide reactions to transfer oxygen atoms. Catalysis by FMOs may proceed through two mechanisms: C4a and N5. The traditional C4a mechanism consists of two half-reactions. In the first half-reaction, the cofactor binds to the enzyme, reducing FAD (or FMN) through two electron transfers. In the second half-reaction, molecular oxygen rapidly reacts with the reduced flavin ($Fl_{red}$), forming a peroxide flavin and oxidizing the substrate [6]. In addition, recently, Teufel and colleagues revealed the N5 redox mechanism involving the bacterial protein Encm [7]. In Encm, histidine 78 is covalently bonded to FAD. This mechanism enables the protonation of the N5 hydroxyl group and hydroxylamine tautomerization in the absence of a cofactor, thereby achieving the hydroxylation of the substrate.

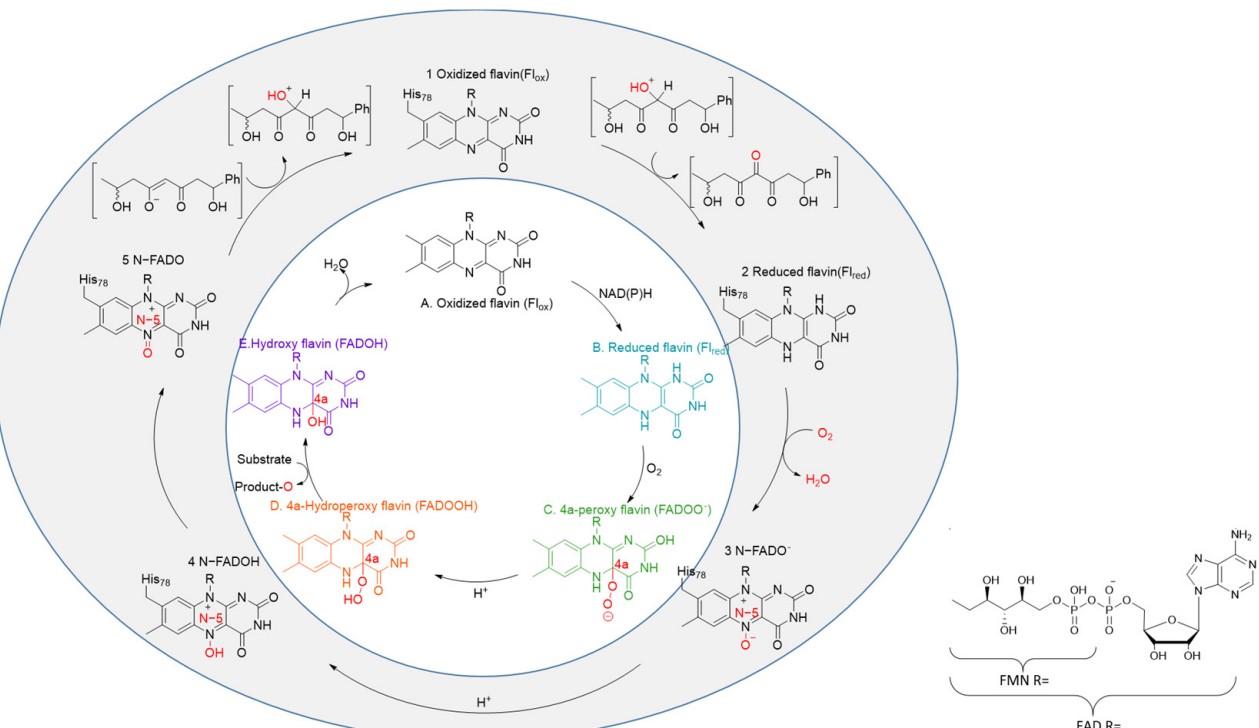

**Figure 1.** The two catalytic mechanisms of FMOs: the C4a mechanism, labeled with letters (in the white area), and the N5 mechanism, labeled with numbers (in the gray area). In the C4a catalytic mechanism, the cofactor NAD(P)H acts as the hydrogen donor, whereas, in the N5 mechanism, the substrate itself serves as the hydrogen donor.

Flavin-dependent monooxygenases (FMOs), as key biocatalysts, exhibit tremendous potential in multiple disciplines, including drug metabolism [4], degradation of harmful substances [8], and synthesis of bioactive molecules [6]. Importantly, unlike cytochrome P450 enzymes (CYPs), FMOs are self-sufficient, not requiring additional redox partner proteins for activation [9]. Moreover, compared to CYPs, FMOs demonstrate higher cat-

alytic efficiency and lower level of substrate inhibition [10,11]. In recent years, significant progress has been made in research on FMO application. This work will discuss the novel applications of FMOs in various fields such as biomedical science, environmental science, and synthetic biology.

## 2. Applications of FMOs in the Biomedical Field

### 2.1. Applications of FMOs in Antibiotic Research

With the advancement of bacterial genomic sequencing and genome mining, the identification of biosynthetic gene clusters for bioactive natural products, such as tumor-inhibiting and antibiotic agents, has accelerated [12]. In the reconstruction of these biological pathways, many new FMOs have been discovered. These FMOs are involved not only in the synthesis of natural products but also in initiating the biodegradation metabolism of various chemicals [13,14]. Additionally, this activity enables bacteria to degrade or deactivate antibiotics. MabTetX, an FMO found in *Mycobacterium abscessus*, belongs to the tetracycline destructase (TetX) family [15]. Recent studies associated MabTetX with tetracycline resistance [16]. MabTetX catalyzes the modification of tetracycline and doxycycline through oxidation reactions. Studies also showed that deletion of the MabTetX gene MAD_1496c increases the bacterial sensitivity to tetracycline and doxycycline. Anhydrotetracycline (ATc), a degradation product of tetracycline, despite its lower antibiotic activity, can effectively inhibit MabTetX, thereby reducing the minimum inhibitory concentrations (MICs) of tetracycline and doxycycline and potentially enhancing their effectiveness against *Mycobacterium abscessus* [16].

Sulfonamide drugs, synthetic antimicrobial agents, are widely used in human and veterinary medicine [17]. However, sulfonamides are chemically difficult to degrade, and the enzymes involved in their degradation are considered to play a role in potential resistance mechanisms [18]. In the bacterium *Microbacterium* sp. CJ77, the enzymes SadA and SadC are involved in the degradation of sulfonamides (Figure 2A, **1**) and are associated with resistance. SadA, a $FMNH_2$-dependent monooxygenase, is responsible for the initial ipso-hydroxylation reaction of sulfonamides (**2**), while SadC, a flavin reductase, participates in the subsequent breakdown of these drugs [19]. Co-expression of this two-component system in *Escherichia coli* was shown to decrease the microorganism's sensitivity to sulfamethoxazole.

**Figure 2.** (**A**) Involvement of SadA/SadC in the degradation of sulfonamides. (**B**) Actinorhodin biosynthesis pathway; ActVA-5/ActVB is involved in two consecutive hydroxylation reactions.

FMOs are also extensively involved in the degradation of aromatic compounds in the environment and in the biosynthesis of secondary metabolites [4]. Recent studies identified the ActVA-ORF5/ActVB system as being involved in the synthesis of the quinone antibiotic actinorhodin (Figure 2B **9**). This system specifically participates in two consecutive hydroxylation reactions (**7**,**8**), which are crucial steps in the biosynthetic pathway of actinorhodin [20].

## 2.2. Applications of FMOs as Pharmaceutical Targets

Kynurenine 3-monooxygenase (KMO), a key enzyme in the Class A FMO pathway operative in the mitochondrial outer membrane, is responsible for catalyzing the hydroxylation of *L*-kynurenine (Figure 3A **10**), thereby producing the neurotoxin 3-hydroxykynurenine (**11**) [21]. Studies indicate that KMO facilitates cancer progression and serves as a prognostic marker for human hepatocellular carcinoma and canine mammary tumors [22]. For instance, in a recent study on triple-negative breast cancer (TNBC), KMO amplification was associated with poorer survival rates. KMO expression is significantly higher in TNBC tumors compared to adjacent normal mammary tissues. In vitro experiments showed that increased KMO expression enhanced TNBC cell growth, colony and mammosphere formation, as well as migration, invasion, and the expression of mesenchymal markers. KMO also upregulates the expression and promoter activity of pluripotency genes. It regulates pluripotency genes through β-catenin, playing an oncogenic role in the progression of TNBC [23]. Currently, research is focusing on developing effective KMO inhibitors (Figure 3B), primarily through high-throughput screening [24] and drug design [25].

**Figure 3.** (**A**) KMO catalyzes the hydroxylation of *L*-kynurenine. (**B**) Potential KMO inhibitors [24]. (**C**) Three SQLE inhibitors (NB-598, $IC_{50}$ = 63 nM, Cmpd-4, $IC_{50}$ = 69 nM, terbinafine, $IC_{50}$ = 7.7 μM).

Human squalene epoxidase (SQLE), a key enzyme in cholesterol biosynthesis, has attracted research interest in recent years. NB-598 (Figure 3C **14**) and Cmpd-4 (**15**) were developed as potent SQLE inhibitors, with $IC_{50}$ values of 63 nM and 69 nM, respectively. In contrast, the common antifungal drug terbinafine (**16**), although sharing a similar tertiary amine structure with these inhibitors, possesses a larger naphthyl group. This structural difference corresponds with its weaker inhibitory effect ($IC_{50}$ of 7.7 μM) [26]. High-resolution crystallographic analysis also revealed the binding sites of these inhibitors [27]. The interaction between the Tyr195 residue and the amine group of the inhibitors may be crucial in designing the next generation of SQLE inhibitors.

## 2.3. Applications of FMOs in Drug Synthesis

HpaBC, a bifunctional flavin-dependent monooxygenase in *Pseudomonas aeruginosa*, is involved in catalyzing the hydroxylation of cinnamic acid derivatives [28]. HpaBC demonstrated high activity in the hydroxylation of 3-(4-hydroxyphenyl)-propanoic acid (Figure 4A **17**), producing 3-(3,4,5-trihydroxyphenyl)-propanoic acid (**19**), a compound with

significant anti-proliferative effects in human cancer cells and high medicinal potential [29]. Recent studies showed that HpaBC can also hydroxylate *L*-tyrosine (**20**) to produce *L*-DOPA (**21**), a prescription medication for Parkinson's disease [30]. The characterization of HpaBC's function and structure revealed its ability to hydroxylate a range of phenolic substrates, including tyrosol, hydroxyphenyllactic acid, coumaric acid, hydroxybenzoic acid, and phenol [31]. Further enzyme engineering, by substituting selected HpaBC residues with homologous residues from *Pseudomonas aeruginosa*, revealed that these mutants possess highly region-specific aromatic hydroxylation capabilities [32].

**Figure 4.** (**A**) HpaBC-catalyzed hydroxylation of 3-(4-hydroxyphenyl)-propanoic acid and *L*-tyrosine. (**B**) Stereochemical selectivity in NotI/NotI'-catalyzed hydroxylation.

Fungal indole alkaloids are a class of bioactive natural products including notoamide, which exhibits notable anticancer activity [33]. In the synthesis of these compounds, the flavin-dependent monooxygenases NotI and NotI' (sharing 85% sequence homology) catalyze the rearrangement of the semi-indole to form the spiro-indole portion. Interestingly, both oxygenases can process any enantiomer of stephacidin A (Figure 4B **22**,**24**), leading to the corresponding notoamide isomers (**23**,**25**). Notably, (−)-stephacidin A (**24**) is more reactive than (+)-stephacidin A (**22**), which might be attributed to the conversion of (+)-stephacidin (**22**) A as an evolutionary process [33].

## 3. Applications of FMOs in the Biomedical Field

### 3.1. Application of FMOs in the Processing of Pesticides

The pesticide degradation process involves several key steps, including the cleavage of functional groups, the hydroxylation of aromatic heterocycles, and the further breakdown of hydroxylated aromatic heterocycles [34]. In this degradation pathway, the cleavage of functional groups is the initial step, primarily involving the release of heteroatoms, through reactions such as dealkylation, decarboxylation, and dehalogenation. For instance, the bifunctional FMO TomAB in *Acidovorax* sp. is involved in the cleavage of the C-S bond in the thiocarbamate herbicide thiobencard (Figure 5A **26**) [35]. Chlorpyrifos, one of the most commonly used organophosphate insecticides, is typically degraded in the environment to 3,5,6-trichloro-2-pyridinol (TC2P, **29**) [36]. TcpA from the X1$^T$ strain can progressively dechlorinate TC2P in the presence of the redox partner protein (Fre) [37]. Additionally, the OdcA gene in a *Pigmentiphaga* sp. strain, identified during the study of the herbicide bromoxynil's hydrolysis product 3,5-dibromo-4-hydroxybenzoate (DBHB, **31**), exhibits decarboxylation activity, converting DBHB to 2,6-dibromohydroquinone (**32**) in the presence of cofactors. However, the activity of OdcA significantly decreases with monobrominated analogs [38].

**Figure 5.** (**A**) Applications of TomAB, TcpA, and OdcA in functional group cleavage. (**B**) Applications of CfdCX, MeaXY, DcmB1B2, NphA1A2, and CehC1C2 in aromatic ring hydroxylation, with DcmB1B2 and CehC1C2 catalyzing the formation of vicinal diol structures.

Aromatic hydroxylation, especially ortho-dihydroxylation, produces dihydroxylated compounds crucial for ring cleavage processes [34]. The bifunctional flavin-dependent monooxygenases CfdC$_{CDS-1}$ and CdfX from *Sphingomonas* spp. are capable of hydroxylating furan phenol (Figure 5B **33**), a hydrolysis product of furan [39]. In *Sphingobium baderi* DE-13, the bifunctional FMO MeaXY can completely degrade 2-methyl-6-ethyl aniline (MEA, **35**) and hydroxylate MEA and its metabolic product 2,6-diethyl aniline (EDA, **36**) to produce para-hydroxylated amino derivatives. MeaXY is also highly conserved in other *Sphingomonas* strains degrading MEA [40]. Another bifunctional flavin-dependent monooxygenase, DcmB1B2, can ortho-hydroxylate 4-chlorophenol to form 4-chlorocatechol (**37**) [41]. Similarly, NphA1A2 (56–79% sequence similarity to DcmB1B2) can convert 4-nitrophenol (**39**) to 4-nitrocatechol (**40**) [42]. In Rhizobium sp. X9, the bifunctional FMO CehC1C2, identified in the typical degradation pathway of the carbamate pesticide carbaryl, catalyzes the conversion of 1-naphthol (**41**) to 1,2-dihydroxynaphthalene (**42**) [43]. CehC1C2 can utilize both FAD and FMN as cofactors, with FAD showing higher catalytic activity (Km and kcat/Km values of 74.71 ± 16.07 μM and (8.29 ± 2.44) × $10^{-4}$ $s^{-1} \cdot μM^{-1}$).

### 3.2. Application of FMOs in Plant Life Activities

In recent years, studies in mammals, bacteria, and fungi significantly advanced research on flavin-dependent monooxygenases (FMOs), particularly in areas closely related to human health and biotechnological applications [44–46]. In contrast, although plants possess a greater diversity of FMOs than other organisms, the endogenous substrates of plant FMOs remain largely unidentified, which considerably hampers research in this area [47,48].

Plant FMOs are classified as Class B FMOs, characterized by three conserved sequences: the FMO recognition motif (FXGXXXHXXXY/F), the FAD binding motif (GXGXXG), and the NAD(P)H binding motif (GXGXXG) [49]. The FMO recognition motif is located at the binding pocket junction, ensuring proper structural rotation and conformational changes [50]. The primary subclass of plant FMOs is BFMO, but recent studies on BVMOs from mosses have expanded our understanding of the diversity of plant FMOs [51].

Cm-BVMO from *Cyanidioschyzon merolae* (Cm) and Pp-BVMO from *Physcomitrella patens* (Pp) are currently the only reported plant-derived BVMOs. Cm-BVMO is known as the most thermostable Type I BVMO, with an apparent melting temperature of 56 °C. Pp-BVMO was demonstrated to insert oxygen atoms into C-C bonds, catalyzing the conversion of phenyl-lacetone (Figure 6 **43**) to benzyl acetate (**44**). Interestingly, mutating the Y residue in the FxGxxxYxxxWP motif to an H residue in the enzyme sequence resulted in higher activity on most tested substrates [51]. However, the physiological role of Pp-BVMO remains unclear. The

recombinant enzyme exhibits relatively low kcat values (in the 0.1–0.2 s$^{-1}$ range), suggesting a potential role in the secondary metabolism of these photosynthetic organisms.

**Figure 6.** The involvement of FMOs in plant life catalytic metabolism.

BFMO is the largest subclass of plant FMOs, typically catalyzing oxidation reactions that lead to the hydroxylation of heteroatoms like nitrogen and sulfur [4,52].YUCCA, the first functionally characterized FMO in plants, is involved in the synthesis of the hormone auxin (IAA, **46**) [53]. In Arabidopsis, a series of YUCCA enzymes (YUC 2-11) were identified [54–56], and other plant species also possess complete YUCCA genes [57,58]. Numerous YUCCAs are involved in catalyzing the oxidative decarboxylation of IPA (**45**) to IAA (**46**), as well as the oxidative decarboxylation of phenylpyruvic acid to produce phenylacetic acid [59] and the *N*-hydroxylation of TAM (**47**) to produce *N*-hydroxy TAM (**48**) [53]. YUC6, the most extensively studied enzyme in this family, not only is involved in auxin synthesis but also exhibits reductase partner enzyme activity, enhancing peroxidase activity and the scavenging of reactive oxygen species [60]. The overexpression of YUC6 enhances plant drought tolerance and reduces leaf senescence [61]. Despite significant progress in studying YUCCA functions, the catalytic mechanism of these enzymes remains largely elusive.

The AsFMO1 enzyme in garlic shows high sequence similarity to the *S*-methyl methanethiosulfinate *S*-oxygenase (FMOG*S*-OX) enzyme in Arabidopsis [62,63]. AsFMO1 exhibits high stereoselective *S*-oxygenation activity towards (+)-alliin [(RCSS)-*S*-allylcysteine sulfoxide] (**49**), with an apparent Km value of 0.25 mM, resulting in the formation of allicin (**50**). This compound provides the distinctive flavor of the Allium genus and is beneficial for human health [62].

Plants can induce a broad-spectrum immune response to pathogen infections, known as systemic acquired resistance (SAR). In Arabidopsis, AtFMO1 plays a critical role in inducing and regulating SAR, crucial for disease resistance and plant immunity [64]. As a piperidine *N*-hydroxylase, AtFMO1 catalyzes the conversion of the lysine catabolite pipecolic acid (Pip, **53**) into *N*-hydroxypipecolic acid (NHP, **54**). The exogenous application of AtFMO1 can overcome the FMO deficiencies caused by a lack of NHP, thereby enhancing immunity against oomycete infections [65].

## 4. Applications of FMOs in Natural Product Synthesis

Natural products (NPs) typically do not partake in primary metabolic processes but are encoded and synthesized through secondary metabolic pathways. They possess unique structures and play a crucial role in drug discovery [66–68]. By deciphering and reconstruct-

ing metabolic pathways in organisms, a variety of naturally occurring compounds with unique bioactivities can be produced in cell factories, such as medicinal terpenes [69–71], polyphenols [72–74], and alkaloids [73,75,76]. The catalytic reactions of FMOs in synthesizing natural products include hydroxylation, epoxidation, Baeyer–Villiger oxidation, decarboxylation, dehalogenation, and dealkylation [52,77].

### 4.1. Application of FMOs in the Biosynthesis of Polyether through Epoxidations

Natural polycyclic polyethers are primarily composed of acetate, propionate, and butyrate units, typically containing multiple chiral centers [78]. Their unique multi-furan ring structure facilitates the transport of metal ions. In polyether carriers, metal ions often chelate with oxygen atoms, which enhances their transfer across biological membranes carried out by the hydrophobic carriers, as well as their antibiotic activity [79]. For instance, monensin A (Figure 7A **57**), a typical ion carrier polycyclic polyether antibiotic, disrupts the ion concentration gradient across cell membranes by chelating ions like $Na^+$ and $K^+$, widely used in the treatment of coccidiosis in poultry and cattle [80]. In the biosynthetic pathway of monensin A, the flavin-dependent epoxidase MonCI and epoxide hydrolases MonBI/BII were discovered. MonCI is involved in the stereoselective epoxidation of three double bonds in the precursor substrate [3,81], while MonBI/BII participate in epoxide ring-opening cascade reactions to form new five- and six-membered rings. MonCI catalyzes the triple epoxidation of premonensin A (**55**) to form (12*R*, 13*R*, 16*R*, 17*R*, 20*S*, 21*S*)-triiepoxypremonenin (**56**), with each epoxidation occurring in a highly stereospecific manner. This process is facilitated by the unusually large substrate binding cavity of MonCI, accommodating various conformations of premonensin A. This capability of performing multiple stereospecific epoxidations demonstrates the extraordinary functionality and vast potential of FMOs [3].

**Figure 7.** (**A**) Biosynthesis of monensin A, involving MonCI. (**B**) Biosynthesis of lasalocid A, involving Lsd18.

Lasalocid A (**60**) is one of the simplest polycyclic polyether ion carriers, similar to monensin, composed of a tetrahydrofuran ring and a tetrahydropyran ring. A key enzyme in the synthesis of lasalocid A is Lsd18, a flavin-dependent epoxidase [82].Lsd18 catalyzes the conversion of prelasalocid A (Figure 7B **58**) to bisepoxyprelasalocid A (**59**), an essential step in the formation of the polycyclic polyether compound. Subsequently, Lsd19 is involved

in the epoxide hydrolysis reaction to form the final polycyclic polyether compound. The entire process is carried out in a stereoselective manner.

## 4.2. Application of FMOs in the Biosynthesis of Natural Products through Dearomatization

In natural product synthesis, FMOs participate in the oxidative dearomatization of various substances. Compared to traditional metal catalysts like I[III], I[V], Pb[IV], and C[I]), FMOs offer unique stereoselectivity and precise site selectivity, reducing metal byproduct contamination and improving atom economy [83,84]. Tropolones are a class of seven-membered ring natural products, characterized by a core structure containing a ketone and a hydroxyl group [85]. Tropolones and their derivatives display multiple activities in biology and medicine, with puberulic acid (5-hydroxy stipitatic acid) showing potent anti-malarial activity ($IC_{50}$ = 10 ng·mL$^{-1}$) [86]. TropB, a flavin-dependent monooxygenase, participates in the pathway leading to the natural product stipitatonic acid, selectively hydroxylating 3-methyl-octanal (Figure 8 **61**) at the C-3 position for dearomatization [85]. Enzymes with similar mechanisms include AzaH and AfoD. AzaH, from a silent *Aspergillus niger* gene cluster, participates in the synthesis of azanigerone A (**66**), inducing an "*R*" configuration at the newly formed stereocenter [87]. AfoD is involved in asperfuranone (**69**) synthesis, producing a complementary "*S*" configuration [88]. SorbC, involved in sorbicillactone A (**72**) synthesis, differs in regio selectivity from AzaH, AfoD, and TropB [89]. Additionally, the flavin-dependent monooxygenase TerC, encoded by the *terCDEF* gene, catalyzes the dearomatization of 6-hydroxymellein (6-HM, **73**) to form 1,4-benzoquinone (**75**). This reaction, controlled solely by C-7 substitution, triggers a skeleton alteration through a bifurcated reaction cascade, forming benzoquinone or pyrone structures and offering a novel approach for the synthesis of 1,4-benzoquinone [90].

**Figure 8.** Flavin-dependent monooxygenases TropB, AzaH, and AfoD catalyze a hydroxylation reaction at the C-3 position. SorbC catalyzes a hydroxylation reaction at the C-5 position. In parentheses are the final natural products synthesized in the respective pathways.

### 4.3. Application of Baeyer–Villiger Monooxygenases in Natural Product Synthesis

Baeyer–Villiger monooxygenases (BVMOs) are a class of flavin-dependent monooxygenases that catalyze the Baeyer–Villiger (BV) oxidation of ketones and cyclic ketones into esters or lactones by inserting an oxygen atom near the carbonyl group, in the presence of cofactors. BVMOs can also oxidize heteroatoms, as occurs in *N*-oxidation or sulfoxidation [91]. Compared to metal catalysts, BVMOs exhibit superior regioselectivity and enantioselectivity and can process a wide range of substrates, including cyclic, substituted cyclic, aromatic, and linear ketones, aldehydes, bicyclic ketones, and various steroids [92]. BVMOs are categorized into two types based on the flavin cofactor used: Type I BVMOs contain oxidative and reductive domains with two dinucleotide motifs (Rossmann fold) for binding FAD and NAD(P)H [93,94], while Type II BVMOs consist of two different peptide components, i.e., one oxidative component binding FMN as a cofactor, and another reductive component utilizing NADH as a cofactor [95].

Abyssomicins/neoabyssomicins, isolated from *Verrucosispora* and *Streptomyces* species, are a class of anti-infective spirane lactone antibiotics [96–98]. AbmE2/AbmZ is a bifunctional BVMO involved in the catalytic conversion of abyssomicin 2 (Figure 9A **76**) to neoabyssomicin B (**77**), categorized as a Type II BVMO. Interestingly, abyssomicin 2 (**76**) exhibits antibacterial activity against Gram-positive pathogens, including clinically methicillin-resistant Staphylococcus aureus (MRSA), with MIC values of 3–15 µg/mL, while its derivative neoabyssomicin B (**77**) does not show such activity. This suggests that AbmE2/AbmZ may act as a resistance gene, activating a self-defense strategy through the transformation of toxic substances [99].

**Figure 9.** (**A**) AbmE2/AbmZ catalyzes the formation of neoabyssomicin B, devoid of antibacterial activity, from abyssomicin 2. (**B**) The application of BoBVMO and CbBVMO in acetone synthesis. (**C**) The catalytic application of PaBVMO and PpBVMO on linear fatty ketones. PaBVMO exhibits stereochemical selectivity (**83**:**84** = 71:29), while GsBVMO demonstrates stereochemical selectivity (**83**:**84** = 3:97); the numbers in parentheses indicate the corresponding hydrolysis products. (**D**) The applications of PAMO and its mutants in catalysis.

Optically active sulfoxide structures are extensively used in triazole drugs, such as the chiral proton pump inhibitors dexlansoprazole (Dexilant™, the R-enantiomer of lansoprazole) [100], dexrabeprazole (Dexpure™, the R-enantiomer of rabeprazole), and esomeprazole (Nexium™, the *S*-enantiomer of omeprazole) [101], with esomeprazole widely used in the clinical treatment of gastrointestinal disorders [102]. BoBVMO is the first discovered natural enzyme capable of catalyzing the asymmetric sulfoxidation of bulky prazole thioethers, showing its highest activity toward benzyl methyl sulfide (specific activity of 0.117 U/mg) and only a modest activity of 0.69 mU/mg toward lansoprazole sulfide (LPS, precursors for lansoprazole, **80**) [103]. Through genomic mining of BoBVMO, CbBVMO from Cupriavidus basilensis (62% sequence similarity) was identified, exhibiting a higher specific activity of 39 mU/mg toward LPS (Figure 9B **81**), completing the full conversion of 10 mM LPS in 35 h, and showing good activity toward several other prazole sulfides. Overall, the catalytic efficiency of CbBVMO remains suboptimal. This issue might be addressed adequately through protein engineering and reaction engineering [104].

In biobased chemicals, ω-hydroxy fatty acids (C8–C14) are significant functional compounds, widely used in fragrances, preservatives, adhesives, and pharmaceutical intermediates. These compounds are characterized by having both carboxylic and hydroxyl functional groups at the opposite ends of the fatty acid chain [105–107]. Previous studies showed that PpBVMO from *Pseudomonas putida* KT2440 catalyzes the insertion of oxygen atoms at high substitution sites in asymmetric linear ketones, followed by hydrolysis to produce the corresponding ω-hydroxy fatty acids [108]. Similarly, PfBVMO from *Pseudomonas fluorescens* DSM 50106 can catalyze the formation of α,ω-dicarboxylic acids [109]. Based on this research, PaBVMO from *Pseudomonas aeruginosa* demonstrated higher regioselectivity than PfBVMO, especially for long-chain (C16–C19) ketones, producing up to 95% dicarboxylic monoesters [110]. Recent studies focused on enhancing the selectivity for monocarboxylic esters (PpBVMO **83:84** = 26:74) (Figure 9C), as demonstrated by GsBVMO from *G. sihwensis*, with 54% sequence similarity to PpBVMO, exhibiting high selectivity and activity in catalyzing the conversion of medium-to-long-chain ketone acids into monocarboxylic esters. Surprisingly, enzyme engineering studies on GsBVMO revealed that the mutant GsBVMO$_{C308L}$ is an efficient biocatalyst, effectively converting 10-ketostearic acid into 9-nonyloxy nonanoic acid (60.5 gL$^{-1}$d$^{-1}$) [111].

It was reported that the use of the natural indigo dye dates back over 4000 years, primarily in the textile industry. In Asia, this and similar compounds are also used as therapeutic agents for various diseases [112]. PAMO is a Baeyer–Villiger monooxygenase (BVMO) from the thermophilic actinomycete *Thermobifida fusca*, capable of catalyzing the conversion of phenylacetone to lactone, with a catalytic constant (kcat) of 1.9 s$^{-1}$ and a Michaelis constant (Km) of 59 μM (Figure 9D **86**) [113]. Research showed that the optimized mutant B-PV demonstrates remarkable efficiency in the conversion of cyclohexanone, achieving a 90% conversion rate within 7 h [114]. Iterative saturation mutagenesis of PAMO led to two mutants: PAMO$_{HPCD}$ and PAMO$_{HPED}$ [115]. Compared to the wild-type molecule, these proteins underwent seven mutations at the amino acid positions 93–94 and 440–444. Notably, in PAMO$_{HPCD}$, the leucine residue at position 443 was mutated to cysteine (C), while in PAMO$_{HPED}$, it was mutated to glutamate (E). When expressed in *Escherichia coli*, these mutants catalyze the hydroxylation of the tryptophan lyase product indole (**87**), which subsequently reacts with the indole oxidant isatin (**89**) to form indigo (**90**) and indirubin (**91**). The catalytic efficiencies are kcat 5.4 s$^{-1}$, Km 58.2 μM for PAMO$_{HPCD}$ and kcat 1.7 s$^{-1}$, Km 109.0 μM for PAMO$_{HPED}$. Additionally, these mutants also exhibit catalytic activity towards halogenated indoles [115].

## 5. Summary and Outlook

Flavin-dependent monooxygenases (FMOs) play a significant role in the field of biocatalysis, exhibiting tremendous potential for future biosynthesis and drug development. These enzymes utilize flavin adenine dinucleotide (FAD) and flavin mononucleotide (FMN) as cofactors to catalyze various oxidation reactions, including heteroatom hydroxylation,

Baeyer–Villiger oxidation, and epoxidation. FMOs are known for their high regio- and stereoselectivity, crucial in the pharmaceutical industry [6]. The uniqueness of FMOs lies in their ability to catalyze diverse reactions and in their high efficiency. By introducing oxygen atoms through redox processes, they may impart biological activity to substrates, a key aspect in drug discovery and design. Additionally, the inactivation of FMOs can serve as a target for identifying small molecules involved in this process, detecting potential diseases, treating microbial infections, and developing specific inhibitors against their catalytic conformations [77] (Table 1). However, current research on FMOs as drug targets faces significant challenges, as the structures of many key metabolic FMOs remain unresolved. Identifying the structures of the FMOs of interest may become a focal research topic of future biomedicine studies.

**Table 1.** Summary of the flavin-dependent monooxygenases described in this study.

| Enzyme Name | Source Organism | Catalytic Function |
|---|---|---|
| MabTetX | *Mycobacterium abscessus* | Catalyzes the modification of tetracycline and doxycycline |
| SadA | *Microbacterium* sp. CJ77 | Responsible for the initial ipso-hydroxylation reaction of sulfonamides |
| KMO | Human | Catalyzes the hydroxylation of *L*-kynurenine, producing 3-hydroxykynurenine |
| SQLE | Human | Potential drug targets |
| HpaBC | *Pseudomonas aeruginosa* | Catalyzes the hydroxylation of cinnamic acid derivatives |
| TomAB | *Acidovorax* sp. | Involved in the C-S bond cleavage of the thiocarbamate herbicide thiobencard |
| TcpA | X1$^{T}$ strain | Progressively dechlorinates TC2P in the presence of a redox partner protein (Fre) |
| OdcA | *Pigmentiphaga* sp. | Exhibits decarboxylation activity, converting DBHB to 2,6-dibromohydroquinone |
| CfdC$_{CDS-1}$ | *Sphingomonas* sp. | Capable of hydroxylating furan phenol, a hydrolysis product of furan |
| MeaXY | *Sphingobium* baderi *DE-13* | Degrades 2-methyl-6-ethyl aniline and hydroxylates MEA and its metabolic product EDA |
| DcmB1B2 | *Brevundimonas* sp. JT-9 | Ortho-hydroxylates 4-chlorophenol to form 4-chlorocatechol |
| NphA1A2 | *Rhodococcus* sp | Converts 4-nitrophenol to 4-nitrocatechol |
| CehC1C2 | *Rhizobium* sp. X9 | Catalyzes the conversion of 1-naphthol to 1,2-dihydroxynaphthalene |
| Pp-BVMO | *Physcomitrella patens* (Pp*)* | Inserts oxygen atoms into C-C bonds, catalyzing the conversion of phenylacetone to benzyl acetate |
| YUCCA | *Arabidopsis* | Involved in the synthesis of the hormone auxin (IAA) |
| AsFMO1 | *Garlic* | Exhibits high stereoselective *S*-oxygenation activity towards (+)-alliin |
| MonCI | *Streptomyces cinnamonensis* | Involved in the stereoselective epoxidation of three double bonds in premonensin A |
| Lsd18 | *Streptomyces lasalocidi* | Catalyzes the conversion of prelasalocid A to bisepoxyprelasalocid A |
| TropB | *Ttopolone* | Selectively hydroxylates 3-methyl-octanal at the C-3 position for dearomatization in stipitatonic acid synthesis |
| AzaH | silent *Aspergillus niger* | Participates in the synthesis of azanigerone A, inducing an 'R' configuration at the newly formed stereocenter |
| AfoD | *Aspergillus nidulans* | Involved in asperfuranone synthesis, producing a complementary 'S' configuration |

Table 1. *Cont.*

| Enzyme Name | Source Organism | Catalytic Function |
|---|---|---|
| SorbC | *Penicillium chrysogenum* E01-10/3 | Involved in sorbicillactone A synthesis, differs in site selectivity from AzaH, AfoD, and TropB |
| TerC | *terCDEF* | Catalyzes the dearomatization of 6-hydroxymellein to form 1,4-benzoquinone |
| AbmE2/AbmZ | *Verrucosispora* and *Streptomyces species* | Involved in the catalytic conversion of abyssomicin 2 to neoabyssomicin B |
| BoBVMO | *Bradyrhizobium oligotrophicum* | Catalyzes the asymmetric sulfoxidation of bulky prazole thioethers |
| CbBVMO | *Cupriavidus basilensis* | Exhibits a high specific activity toward lansoprazole sulfide |
| PpBVMO | *Pseudomonas putida* KT2440 | Catalyzes the insertion of oxygen atoms in asymmetric linear ketones |
| PfBVMO | *Pseudomonas fluorescens* DSM 50106 | Catalyze the formation of $\alpha,\omega$-dicarboxylic acids |
| PaBVMO | *Pseudomonas aeruginosa* | Demonstrates high regioselectivity toward long-chain ketones |
| GsBVMO | *G. sihwensis* | Exhibits high selectivity in catalyzing medium-to-long-chain ketone acids |
| PAMO | *Thermobifida fusca* | Catalyzes the conversion of phenylacetone to lactone |
| PAMO$_{HPCD}$/PAMO$_{HPED}$ | *Thermobifida fusca* | Optimized mutant of PAMO, effective in the hydroxylation of indole and in the formation of indigo |

In aerobic microorganisms, flavin-dependent monooxygenases (FMOs) play a leading role in the degradation of pesticides. They "attack" chemically stable pesticide substrates through oxygenation, leading to the release of heteroatoms, the dearomatization of aromatic rings, and ring cleavage during the pesticide degradation process. Although FMOs have demonstrated their diverse catalytic capabilities, further elucidation of additional FMOs is required, along with studies on the combination of multiple enzymes, to develop more convenient, efficient, and safe biodegradation products [34].

Research on plant flavin-dependent monooxygenases (FMOs) remains sparse, as highlighted by Schlaich in 2007, who stated that "without any known substrates for plant FMOs, the field would stall" [47]. Indeed, the case of YUC6 suggests that plant FMOs may provide additional physiological benefits unrelated to their catalytic oxidation functions. This revelation expands the significance of FMOs in the realm of plant metabolism and opens new directions for future agricultural applications, particularly in crop improvement. Although limited biochemical and structural analyses have hindered the development of plant FMO research, it is undeniable that plant FMOs, as an underutilized class of enzymes, hold substantial potential [48].

It is well known that natural products are major sources of various chemical products and pharmaceuticals. Traditional chemical synthesis methods struggle to produce complex natural products with unique scaffolds. In this context, flavin-dependent monooxygenases (FMOs) play a crucial role in synthesis pathways as oxygenating enzymes. Although fewer in number compared to the CYP450 family in catalyzing natural product synthesis [116,117], FMOs offer the unique advantages of self-sufficiency, high efficiency, selectivity, and lower toxicity due to fewer toxic byproducts formed in the catalytic processes [118]. The catalytic mechanism framework of FMOs has been largely established, with the revelation of the N5 mechanism suggesting that some FMOs can complete oxidation reactions in the absence of NAD(P)H [7]. The "IN" state of FAD$_{red}$, as demonstrated by studies on GrhO5 crystal structure, revealed that Group A FMOs can complete the catalytic cycle directly in the presence of substrates and NAD(P)H via FAD [119]. Despite significant advancements in understanding FMO catalytic mechanisms and reaction types, several challenges remain. For instance, how to engineer and design enzymes rationally based on FMO catalytic mechanisms to accommodate a broader range of substrates? How to address the expensive

recycling cost of NAD(P)H? There are a few electrochemical and photochemical alternative cycling methods, but their efficiency is still not optimal [120,121].

So far, there are not many patents for the application of FMOs in the industry. However, we firmly believe that, in the current era of rapid research advancements, with the application of protein crystallization [3], cryo-electron microscopy [122], and artificial intelligence technologies such as AlphaFold [123], more catalytic mechanisms and structures of FMOs will be revealed. With the aid of advanced biotechnologies like computational design and directed evolution, the development and application of FMOs across various fields are set to expand significantly [124]. This will not only advance progress in biocatalysis and drug development involving FMOs but also open more possibilities for exploring novel biochemical pathways and innovative therapeutic approaches.

**Author Contributions:** Y.G. conceptualized the review and wrote the manuscript. X.C. supervised and revised the manuscript. All authors have read and agreed to the published version of the manuscript.

**Funding:** This research was funded by the National Key Research and Development Program of China, grant number 2022YFC2106100, the Natural Science Foundation of China, grant numbers 21807088 and 22377098, Projects of International Cooperation in Shaanxi Province of China, grant number 2023-GHYB-08, the Open Project Program of State Key Laboratory of Cancer Biology, grant number CBSKL2022KF13, the Scholarship Program for Science and Technology Activities of Returned Overseas Scholars, grant number 2022-004, and the Open Funding Project of the State Key Laboratory of Bioreactor Engineering, grant number 2018OPEN05.

**Conflicts of Interest:** The authors declare that they have no conflict of interest.

## Abbreviations

The list of acronyms and their full forms as mentioned in the article.

| Acronym | Full Name |
| --- | --- |
| FMOs | Flavin-dependent monooxygenases |
| FAD | Flavin adenine dinucleotide |
| FMN | Flavin mononucleotide |
| CYPs | Cytochrome P450 enzymes |
| KMO | Kynurenine 3-monooxygenase |
| TNBC | Triple-negative breast cancer |
| SQLE | Squalene epoxidase |
| LPS | Lansoprazole sulfide |
| TC2P | 3,5,6-Trichloro-2-pyridinol |
| DBHB | 3,5-Dibromo-4-hydroxybenzoate |
| MEA | 2-Methyl-6-ethyl aniline |
| EDA | 2,6-Diethyl aniline |
| IPA | Indole-3-pyruvic acid |
| IAA | Indole-3-acetic acid |
| SAR | Systemic acquired resistance |
| NPs | Natural products |
| BVMOs | Baeyer–Villiger monooxygenases |
| MRSA | Methicillin-resistant Staphylococcus aureus |

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
