# Peer review of "Recent Applications of Flavin-Dependent Monooxygenases in Biosynthesis, Pharmaceutical Development, and Environmental Science"

_catalysts, doi:10.3390/catal13121495_

Round 1
Reviewer 1 Report
Comments and Suggestions for Authors
The review manuscript entitled " Recent Applications of Flavin-Dependent Monooxygenases (FMOs) in Biosynthesis, Pharmaceutical Development, and Environmental Science" by Guan and Chen involved the current interesting bibliographical research of the flavin-dependent monooxygenases (FMOs) and their applications in different fields of science.
The introduction is according to the developed theme of the manuscript, and it has updated bibliographical references to support the research.
Also, the manuscript is clear, organize, and easy to follow according to the aim of the authors. In addition, the information they described is supported with clear figures that summarize all the data (interesting Figure 1 relate to the catalytic mechanisms of FMOs, please check the size of the molecules in order to see them appropriately).
Please consider the following questions and suggestions to complete the information:
- Did the authors check the current patent state of art of the FMOs utilization? Is there any application reported as a patent?
- Section 4.3. Some interesting pigment application development through these enzymes you can find in Int. J. Mol. Sci. 2022, 23(20), 12544; https://doi.org/10.3390/ijms232012544, please consider this information.
However, some typo and images mistakes should be corrected such as (in yellow, pdf attached):
- L-kynurenine, L-tyrosine, N-hydroxylation, etc.: L and N should be in italics and check them all throughout the manuscript in different words containing L or N.
- R and S for stereoisomerism: R and S should be in italics. Please check them all throughout the manuscript
- IC50: correction IC50
- Figure 4: the image looks blurry, please improve it.
Furthermore, I encourage the authors to include a table summarizing all the information reported in the present manuscript and to improve the conclusions (“mechanisms and structures related to FMOs will be uncovered”, please clarify and give some examples or insights).
Finally, I would like to invite the authors to add the abbreviation list of words at the end of this manuscript.

Reviewer 2 Report
Comments and Suggestions for Authors
Summary: This manuscripts reviews the role of flavin monooxygenases in drug discovery and development. Synthesis reactions catalyzed by enzymes yield stereospecific products. Specific catalytic reactions are shown for flavin monooxygenase enzymes involved in synthesis and degradation of antibiotics, pharmaceutical drugs, pesticides, and natural products.
Minor comments
1. Figure 1 contains errors.
a. The structure of reduced flavin in the white area (B) is different from the structure of reduced flavin (2) in the gray area.
b. Please add a double bond between carbons 4a and 10a in reduced flavin.
c. The oxygen in reduced flavin (B) has 3 bonds, which is incorrect. This oxygen should not be shown as a hydroxyl.
2. Figure 1 legend. Please explain the structures in brackets.
3. Figure 1 legend. Please cite the publication that is the source of the information in Figure 1.
4. Figure 1 legend. Please explain His78 in the gray area. Is His78 intended to represent a covalent link to a protein? Please name the protein and please cite the publication that identified attachment to His78.
5. Line 62 “Robin and colleagues first revealed the N5 redox mechanism in Encm[7]”. Please change to “Teufel and colleagues first revealed the N5 redox mechanism in bacterial protein Encm [7].”
6. Page 3. Please refer to Figure 2 in the text.
7. Page 3. Reference 18 is cited for degradation of sulfonamide drugs. However, reference 18 is about tetracycline, not sulfonamide. Please cite an appropriate publication.
8. Section 2.2 please refer to Figure 3 in the text. Please explain that (12,13) refers to compounds 12 and 13 in Figure 3.
9. Please refer to Figure 4 in the text.
10. Sections 3.1 and 3.2 have the same title “Application of FMO’s in the Processing of Pesticides”. This title is appropriate for section 3.1, but not for section 3.2.
11. Please refer to Figures 5, 6, 7, and 8 in the text. Please refer to every Figure in the text to guide the reader to the appropriate Figure while reading the text.
12. There is no figure for structures 43 to 54.
13. Both Figure 6 and Figure 8 show structures 61-69.
14. Two figures are labeled Figure 8. The first Figure 8 shows structures 61-75. The second Figure 8 shows structures 76-84.
15. Page 9. “TropB, a flavin-dependent monooxygenase encoded in the fungal natural product Stipitatonic acid” Please clarify this phrase. Perhaps you mean “TropB, a flavin-dependent monooxygenase, participates in the pathway leading to the natural product stipitatonic acid.”
